# Exploring the Molecular Underpinnings of Cancer-Causing Oncohistone Mutants Using Yeast as a Model

**DOI:** 10.3390/jof9121187

**Published:** 2023-12-11

**Authors:** Xinran Zhang, Dorelle V. Fawwal, Jennifer M. Spangle, Anita H. Corbett, Celina Y. Jones

**Affiliations:** 1Department of Biology, Emory University, Atlanta, GA 30322, USA; xzhangs@emory.edu (X.Z.); dory.fawwal@emory.edu (D.V.F.); acorbe2@emory.edu (A.H.C.); 2Department of Radiation Oncology, Emory University School of Medicine, Atlanta, GA 30322, USA; jennifer.spangle@emory.edu; 3Biochemistry, Cell & Developmental Biology Graduate Program, Emory University, Atlanta, GA 30322, USA; 4Winship Cancer Institute, Emory University, Atlanta, GA 30322, USA

**Keywords:** histone, oncohistone, budding yeast, fission yeast, epigenetics, cancer

## Abstract

Understanding the molecular basis of cancer initiation and progression is critical in developing effective treatment strategies. Recently, mutations in genes encoding histone proteins that drive oncogenesis have been identified, converting these essential proteins into “oncohistones”. Understanding how oncohistone mutants, which are commonly single missense mutations, subvert the normal function of histones to drive oncogenesis requires defining the functional consequences of such changes. Histones genes are present in multiple copies in the human genome with 15 genes encoding histone H3 isoforms, the histone for which the majority of oncohistone variants have been analyzed thus far. With so many wildtype histone proteins being expressed simultaneously within the oncohistone, it can be difficult to decipher the precise mechanistic consequences of the mutant protein. In contrast to humans, budding and fission yeast contain only two or three histone H3 genes, respectively. Furthermore, yeast histones share ~90% sequence identity with human H3 protein. Its genetic simplicity and evolutionary conservation make yeast an excellent model for characterizing oncohistones. The power of genetic approaches can also be exploited in yeast models to define cellular signaling pathways that could serve as actionable therapeutic targets. In this review, we focus on the value of yeast models to serve as a discovery tool that can provide mechanistic insights and inform subsequent translational studies in humans.

## 1. Introduction

Cancer is the second leading cause of death in the United States [1]. As a collection of diseases characterized by shared genomic alterations, various cancer types reveal a diversity of mechanisms. Thus, each cancer must be investigated individually to elucidate a detailed mechanistic understanding and enable the design of effective therapeutic strategies. Numerous strategies are employed to define the mechanisms of cancer initiation and progression with the goal of improving patient outcomes. These strategies employ a variety of systems to characterize cancers, from unicellular organisms to computational analyses to animal models. Each of these approaches has strengths and weaknesses, and together, the goal is to enable a full mechanistic and physiological understanding of each cancer type to eventually lead to their eradication. Given the variety of genomic alterations that have been defined as cancer drivers and that display a diversity of mechanisms, defining the fundamental cellular processes that underlie those mechanisms is critical. Key discoveries that define our understanding of cellular growth pathways that can contribute to oncogenesis have been made in simple model organisms such as unicellular budding and fission yeast. For example, the basic principles of the cell cycle were initially defined in budding yeast [2]. Many proteins and processes essential to cellular physiology are shared between budding or fission yeast and humans. Therefore, findings in simple eukaryotic model organisms can be readily translated into mammalian systems.

In this review, we highlight the strengths of utilizing yeast models, with special emphasis on budding yeast, to characterize the molecular basis of cancers caused by mutations that convert critically important histone proteins into oncohistones. We provide a basic introduction to histone function, an overview of what is currently known about oncohistones, some specific benefits of using these organisms to investigate the molecular disruptions driven by histone mutations in cancers, and examples describing how yeast model systems have already contributed to the oncohistone field.

## 2. Histones and Nucleosomes in Humans and Yeast

In eukaryotes, DNA is packaged by highly basic histone proteins into the fundamental repeating chromatin unit called a nucleosome [3]. Nucleosomes are nucleoprotein complexes in which ~147 bp of DNA is wrapped around a histone hetero-octamer consisting of two copies of each core histone protein H2A, H2B, H3, and H4 (Figure 1A) [4,5]. The structure of each histone protein includes a globular domain that interacts with the other histones and associated DNA as well as a flexible N-terminal tail (Figure 1A). Each individual nucleosome is connected by a variable length of linker DNA to form the basic structure of chromatin [6]. The linker histone protein H1 is selectively incorporated into nucleosomes, enabling the repeating nucleosome structures to be further assembled into heterochromatin and compacting the linear DNA into highly condensed chromatin fibers [5,7]. Beyond facilitating the packaging of the genomic material, the structure of chromatin, which largely depends on precise positioning of nucleosomes, dictates the cellular machinery’s access to the genome. Hence, chromatin structure determines the ability of this cellular machinery to modulate a wide variety of processes including DNA replication, transcription, cell fate, cell cycle progression, and DNA damage repair via chromatin-mediated regulation [8]. The dynamic and intricate organization of chromatin plays a fundamental role in governing diverse cellular processes, highlighting the essential role of chromatin in the regulation of genomic functions within eukaryotic cells.

Histones are among the most evolutionarily conserved proteins in animals, plants, and fungi [9]. These proteins are encoded by multigene families that exhibit modest heterogeneity at the protein level [10]. In humans, the core histone proteins H2A, H2B, H3, and H4 are encoded by 29, 23, 15, and 15 individual genes, respectively, to form ten, four, three, and one distinct isoforms (Figure 2A and Appendix A) [11]. Though the histone protein sequences are similar, these isoforms can perform different functions in regulating biological processes through variable expression and genomic location [12]. 

Core histone proteins are highly conserved between fungi and humans. For histone H3, where the majority of the well-characterized oncohistone mutations are located [8], budding yeast *Saccharomyces cerevisiae* histone H3 protein shares 90% of its identity with human H3.3 protein (Figure 2B). Additionally, budding yeast histones H2A, H2B, and H4 share 73%, 67%, and 92% of their identity with their human counterparts, respectively, while *Schizosaccharomyces pombe* fission yeast histones share 78%, 69%, 92% and 91% of their identity with human H2A, H2B, H3, and H4 (Figure 2C and Appendix A) [13]. This level of conservation, together with the shared protein functions, as described below, means that yeast core histones can serve as a model for understanding the subtle differences in various histone isoforms as well as mutant histone function in higher eukaryotes, including humans. The linker histone H1, however, is not well conserved across species, with only 37% of shared sequence identity between humans and budding yeast, while fission yeast lacks a linker histone altogether. *S. cerevisiae* chromatin is also organized by the high mobility group box family protein Hmo1 which performs similar but not fully overlapping roles to histone H1 [14]. While budding and fission yeasts can serve as excellent models for the core histones due to their high level of conservation with mammalian histone proteins, this is not true of histone H1, so we do not focus on the linker histone in this review. 

In *S. cerevisiae*, histones H2A, H2B, H3, and H4 are each encoded by two gene copies clustered at four loci [15]. The variants H2A.Z and centromeric H3 are each encoded by a single gene [16]. Transcription of all core histone genes is cell-cycle regulated, activating expression in late G_1_ so that sufficient building blocks for nucleosomes exist during DNA replication [16]. Budding yeast histone H3 is encoded by *HHT1* and *HHT2*, and H4 is encoded by *HHF1* and *HHF2*. These genes cluster at two loci (*HHT1-HHF1* and *HHT2-HHF2*) which are slightly different in DNA sequence but produce identical H3 and H4 proteins [17,18]. However, *HHT2* and *HHF2* contribute more than 80% of the mRNA transcripts encoding cellular histone H3 and H4. The level of the *HHT1* and the *HHF1* transcripts is unchanged when *HHT2* and *HHF2* genes are deleted [19]. Minor phenotypic changes have been observed in *hht2∆hhf2∆* cells, but no obvious phenotypic defects were observed in *hht1∆hhf1∆* cells [19,20]. The other two clustered loci, *HTA1-HTB1* and *HTA2-HTB2*, encode two slightly different protein copies of histone H2A and H2B [21], yet these two pairs of genes have non-equivalent roles as the deletion of *HTA2* and *HTB2* does not result in significant growth defects, but the deletion of *HTA1* and *HTB1* is lethal to cells unless the remaining *HTA2-HTB2* locus is duplicated [15,22]. Hence, the simplicity of minimal gene copies encoding histone proteins render budding yeast an effective model organism for the comprehensive investigation of histones.

Despite the conservation of histone sequences, there are some differences in nucleosome and chromatin structure between yeast and mammals. Budding yeast nucleosomes are spaced more closely to one another as compared to mammalian nucleosomes, with a nucleosome repeat length of ~165 bp in *S. cerevisiae* [23] as compared to ~170–220 bp in mammals [6]. Additionally, budding yeast nucleosomes may be less stable than those found in metazoans, as *S. cerevisiae* DNA unwinds more rapidly after treatment with heat [24]. The interactions between histones H2A and H2B in budding yeast also appear to be weaker than these interactions in metazoan nucleosomes due to some variations in protein sequence, decreasing overall nucleosome stability [25]. These differences in yeast nucleosomes may contribute to *S. cerevisiae* having an open chromatin structure that is generally transcriptionally active, in contrast to mammalian genomes that have large regions of transcriptionally inactive heterochromatin. While yeast is a helpful model in epigenetics studies for the reasons described above, functional differences exist that should be taken into consideration.

One critical regulatory mechanism of chromatin is histone post-translational modifications (PTMs), which constitute the “histone code”. The histone code, referring to different spatial or sequential combinations of histone PTMs, choreographs functional activities by modulating the state of the chromatin and maintains homeostasis in response to cellular cues [26]. This code is comprised of molecules which are covalently bound to histone proteins and include methylation, acetylation, phosphorylation, ubiquitylation, and ADP-ribosylation, among others [8]. The most well studied PTMs occur at the N-terminal tails [8]. Histone PTMs modulate chromatin-dependent functions not only by directly altering the chromatin conformation and DNA-histone interaction, but also by recruiting other cellular machinery. These epigenetic factors play key roles in chromatin remodeling, transcription regulation, replication, and repair, and some are highlighted below [27]. 

Histones are highly basic proteins with abundant lysine and arginine residues that are subject to various PTMs [28]. Among these PTMs, lysine acetylation, which is a reversible process regulated through the opposing activities of histone acetyltransferases (HATs) and histone deacetylases (HDACs), is a well-characterized histone modification [27,29]. Several primary acetylation sites have been identified, with the majority being located within the unstructured N-terminal tail domain. Examples of such sites include H3K14/K18/K23 and H4K5/K8/K12/K16 [30]. Crucial acetylation sites within the globular domain have also been identified, including the extensively studied H3K56ac, which is involved in regulating processes such as DNA-damage response, transcription, and nucleosome packaging [31,32,33]. The acetylation of lysine residues neutralizes the overall positive charge of histones, consequently attenuating interactions between basic histone proteins and negatively charged DNA [34]. Hence, lysine acetylation is generally associated with biological processes dependent on access to DNA, including transcriptional activation and DNA replication, though exceptions exist as well [26,35,36]. In addition to acetylation, methylation is another well characterized PTM. Lysine methylation marks, including mono-, di-, and tri-methylation, and arginine methylation, including mono- and di-methylation, are critical determinants of chromatin state and are involved in processes including regulating DNA damage response and transcription [37,38,39,40,41,42]. In contrast to histone acetylation, methylation does not alter the net charge of histones [27]. Therefore, the effects of methylation are thought to be complex, as compared to acetylation, and depend on the location within the chromatin landscape and the degree of methylation [43]. For example, H3K4 methylation is generally associated with transcription activation, yet di- and tri-methylation at this site can trigger transcriptional repression when bound by the co-repressive histone deacetylase complex mSin3a-HDAC1 [44,45]. H3K36 methylation, catalyzed by a set of enzymes in humans and exclusively by Set2 in yeasts, is another PTM which modulates diverse cellular processes, including active transcription, DNA damage repair, and splicing, among others [26,46,47,48,49,50,51]. Other well-described methylation sites are H3K9 and H3K27, which are both hallmarks of heterochromatin and transcriptional silencing [52,53]. Additionally, many studies have revealed the regulatory roles of a wide range of other histone PTMs, including, but not limited to, phosphorylation, ubiquitylation, SUMOylation, and ADP-ribosylation [27,54]. The histone code holds profound implications for biology, highlighting the intricate regulation in place for the epigenome. 

## 3. Oncohistones Are Novel, Cancer-Causing Histone Mutants

The discovery that modest changes to histone sequence can cause a variety of cancers and neurological disorders in humans was surprising [8]. Mutations in histone genes were largely dismissed as drivers of disease in earlier epigenetic research because of their genetic redundancy. In humans, histone genes are encoded in clusters resulting in many copies of each histone gene [55]. Focusing on histone H3, which is the protein with the most well-defined variants that contribute to disease, humans have 15 copies of genes encoding H3 variants (Figure 2A). Several other non-canonical H3 encoding genes and isoforms exist (Figure 2A, marked by *), but their contribution to histone biology is either not well defined or is limited by genomic location or cell type specificity. Diploid cells have at least 30 alleles encoding histone H3 protein. A somatic mutation in a single allele was assumed to have a low likelihood of producing a phenotype as 29 other wildtype H3 alleles were present, even though not all of these may express protein. Other epigenetic factors, however, have been implicated in cancers [56,57,58,59,60], highlighting the essential roles that dynamic regulation of epigenetic marks play in normal cellular function. Specifically, genomic alterations in enzymes involved in DNA methylation (e.g., DNMT3A) [56], histone marks (e.g., MLL mutations and translocations and histone deacetylases) [57,60], and chromatin remodelers (e.g., SWI/SNF complex members ARID1A and SMARCA4) were identified in various cancers [58]. Underscoring the importance of regular epigenetic programming to promote normal cell function, “nonmutational epigenetic reprogramming” was recently identified as a new prospective hallmark of cancer [59]. Clearly, the proteins involved in the regulation of chromatin at every level are critical for contributing to normal cell growth.

A paradigm shift occurred when missense mutations in genes encoding histone proteins were linked to multiple cancers, providing evidence that such mutations could convert critical histone proteins into cancer-driving “oncohistones” (Figure 1B). The first oncohistone mutants discovered were H3K27M, H3G34V/L/R/W, and K3K36M [61,62,63]. H3K27M and H3G34R and V were identified in brain tumors such as gliomas and glioblastomas [61,62], while H3G34W and L were discovered in giant cell tumors of bone [63]. H3K36M is present in 95% of chondroblastoma tumors as well as in head and neck squamous cell carcinomas, melanoma, and bladder and colorectal cancers [63,64]. The oncohistone mutants are encountered in higher proportions in pediatric cancer cases as opposed to adults for reasons that are currently unknown. With rare exceptions, established oncohistone mutations occur in two of the fifteen H3 genes, *H3-3A* and *H3-3B*, which encode the H3.3 variant [61,62,63] (Figure 2A). Histone variants H3.1 and H3.2 are replication-dependent, meaning they are only incorporated into chromatin during DNA synthesis. Histone H3.3, however, is replication independent and is deposited throughout the cell cycle, particularly in regions undergoing active transcription and in constitutively heterochromatic regions, such as telomeres [65,66]. The reason for oncohistones’ bias for H3.3 genes is currently not understood, though it may be related to the direct impact this histone variant would have on actively transcribed genes.

The major molecular etiology for oncohistone-driven cancers that have been defined thus far appears to be via the disruption of key PTMs [67,68]. Both H3K36M and the mutants H3G34R/L/V result in reduced methylation at H3K36 [68,69,70,71]. H3K36M acts as a pseudosubstrate for the H3K36me3 methyl transferase SETD2, outcompeting the wildtype lysine residue by binding the enzyme with high affinity, and sequestering SETD2 to prevent deposition of methyl groups on normal H3 proteins [68,70]. This trapping of SETD2 and the reduction in the enzymatic activity of H3K36me1 and H3K36me2 methyl transferases as well inhibits global H3K36 methylation [68,70]. The residues of H3G33-G34 are also bound by SETD2 and contribute to the accurate recognition of the H3K36 target [69,71]. When altered from glycine to arginine, lysine, or valine, these residues impair SETD2 binding to histone H3 and decrease SETD2 enzymatic activity, greatly reducing H3K36 methylation in *cis* [69,71]. The H3G34 mutants also cause increased H3K27me3 and reduced H3K27ac, though this is secondary to the alterations in H3K36 methylation rather than a direct effect on the H3K27 enzymes [71]. Similar to H3K36M, H3K27M acts as a competitive pseudosubstrate inhibitor for the Polycomb Repressive Complex 2 (PRC2), which contains the H3K27 methyltransferase Ezh2, decreasing Ezh2 activity [67,72,73]. Furthermore, the H3G34 mutants lead to disruptions in gene expression profiles that are similar to those found in cancers [74]. In addition to the oncohistone mutants at H3K27, G34, and K36, there are many other undefined mutations in the genes encoding histones that occur at high frequency in cancers [8,75]. The vast majority of these mutants are in residues within the globular domains of the proteins, raising the hypothesis that an additional mode for driving oncogenesis is via disrupted histone-DNA interactions or nucleosome stability [8,75,76]. 

In cells expressing H3K36M or H3K27M, changes in gene expression have been documented [68,77,78,79]. In the case of H3K36M cells, the reduction in H3K36 methylation causes the redistribution of H3K27me3 from primarily genic to intergenic regions, recruiting Polycomb Repressive Complex 1 (PRC1) to these new marks and away from typically repressed genes, derepressing those genes [68,80]. Thus, oncohistones can also cause changes to the global epigenetic landscape. Beyond affecting transcription, many other processes may be impacted by oncohistone changes. H3G34 mutant cells display increased genome-wide mutation frequencies, both in engineered in vitro lines and when comparing whole genome sequencing data sets between H3G34 mutant and non-H3G34 mutant pediatric glioma tumors [69], suggesting that DNA repair defects are specific to H3G34 mutant tumors. Additionally, methylation at H3K36 is an early signal for recruiting factors to the nonhomologous end following a DNA double-strand break [81,82]. Thus, the inhibition of H3K36 methylation in H3K36M cells likely reduces their ability to repair double-strand breaks. Various processes are disrupted by oncohistones, and each may contribute to cancer development. 

In addition to the bona fide oncohistones described above, many other recurrent histone genomic alterations have been associated with cancers. Though, whether they drive oncogenesis has yet to be fully defined. Various mutations identified in the genes encoding the linker histone H1 are primarily associated with lymphomas and are present in other cancer types as well [83,84]. Some of these mutations appear to disrupt the ability of H1 to associate with chromatin [85]. Another histone mutant that clearly improves cellular growth is H2BE76K [76]. This mutation, which lies within the globular domain of histone H2B, destabilizes nucleosome integrity, alters H2B genomic localization, and disrupts gene expression [86]. Further experimentation in a mouse model remains to be carried out to definitively determine the cancer driving ability of these mutations. 

While the mechanisms by which a few bona fide and likely oncohistones may drive oncogenesis have been explored, there are many additional putative oncohistones. The use of genetic model systems such as yeast to understand how the oncohistone missense mutations alter histone function is valuable for developing hypotheses that can then be tested in mammalian cells and explored in vivo. Analyzing mutation frequencies across all H3 genes may be helpful for identifying patterns that support cancer progression. To explore this idea, we examined mutation frequencies in genes encoding histone H3 across all cancer types using publicly available data from 5406 adult patient tumor samples obtained from the COSMIC and cBioPortal databases (Figure 3A). A similar analysis that assessed cancer-associated mutations in histone genes was performed recently, though with a more limited dataset and a broader scope of genes including all core histones and the linker histone [84]. They found that mutations that were recurrent in cancers tended to occur in specific histone genes, many of which were associated with proliferative sequences, and that the residues with the highest mutation frequencies were commonly conserved across species and known sites of PTMs [84]. For the analysis performed here, we focused on the canonical H3 genes and, specifically, the residues that are conserved with *S. cerevisiae* H3. The top ten residues with the highest mutational burden are H3R2, K27, G34, K36, E50, E73, E97, E105, R116, and R131. As expected, the residues altered in known oncohistone mutants (H3K27, G34, and K36) show the greatest number of changes. It is possible that the high number of alterations detected at these residues is due to ascertainment bias, where these sites were assessed specifically rather than in a nonbiased manner in tumors after being acknowledged as oncohistones. The rest of the residues in the top ten are all glutamic acid or arginine, which are both charged amino acids. The previous pan-cancer study also observed that many of the most frequently mutated residues were charged [84]. Altering the charge of these residues has the potential to disrupt interactions between histones and other proteins or DNA, possibly to a greater extent than changes at non-charged residues, which may explain the high frequency of these specific amino acid substitutions. 

The most common type of mutation identified is a premature stop codon, occurring 2905 times across all residues. Limiting individual histone expression in this way may upset the balance in the histone pool and impact histone hetero-octamer formation. When considering missense mutations, certain variants only occur with high frequency at a single residue. For example, the primary amino acid change detected at H3K27 is conversion to methionine (994 of 1215 total missense mutations), the primary change detected at H3G34 is conversion to tryptophan (403 of 438 total missense mutations), and the primary change detected at H3G34 is conversion to arginine (155 of 239 total missense mutations). An additional pattern that emerged from this analysis was that certain amino acids are converted to specific other amino acids with high frequency (>200 instances) but at no particular position within the histone open reading frame (Figure 3B). For example, a common change is glutamic acid to lysine (E > K) (i.e., H3E50K, E59K, E73K, E94K, E97K, E105K, and E133K), switching the charge at this position from positive to negative. In addition, arginine is commonly changed to histidine or cysteine. We also commonly detected a change from alanine to valine, which would maintain the hydrophobic nature of the residue while modestly increasing the size. Each of these recurrent changes that were not position specific can be achieved through a single base change in the original codon, which may contribute to the bias. Functional studies are required to determine whether these amino acid substitutions alter histone function, providing insight into whether the underlying genetic changes are likely to be passenger mutations or oncogenic drivers. Yeast models can provide an invaluable tool to perform such analyses.

## 4. Advantages of Using Yeast Models to Investigate Oncohistones

While much has been uncovered about oncohistones since their discovery about a decade ago [61,62,63], many outstanding questions remain fully understand how the conversion from essential protein to oncogenic driver occurs. A major complication in the investigation of how an oncohistone alters histone function and growth patterns in human cells is the large number of histone genes. Engineering cell lines that faithfully recapitulate the genetic environment found in disease or that solely express mutant histone in the absence of wild type histone is quite technically challenging. While one recent study in mouse embryonic stem cells introduced an H3K27 mutant (H3K27R) into all 28 alleles of the genes encoding histone H3 [87], the engineering required for designing and generating the lines is not conducive to screening a large number of potential oncohistone mutations for functional changes. Therefore, organisms such as budding or fission yeast can circumvent the challenges posed by mammalian cell lines by having fewer histone genes and allow modeling of oncohistone mutations in a simpler genetic context where the histone variant can be readily expressed as the sole histone protein present in the cell. 

As the majority of oncohistone studies have focused on histone H3, focusing on this core histone protein is of particular interest. Notably, in comparing yeast and human histone H3, the only residues in the N-terminal tail that are not strictly identical are conservative changes (Figure 2B), meaning that the amino acids likely serve similar biochemical functions. This observation is crucial as the N-terminal tail is critical for the dynamic receipt and sending of epigenetic signals via the histone code. Not only is the protein sequence highly similar, but the 3-dimensional structures of human and yeast H3 proteins are also highly conserved [25]. These similarities suggest that histone H3 performs analogous functions in each species, and thus, that molecular disruptions caused by the oncohistone mutants would be similar. 

Importantly, many associated epigenetic factors are also highly conserved between humans and budding yeast [88,89,90]. Given the dynamic and layered regulation that occurs in histones, findings uncovered in model organisms are greatly strengthened by the shared structures and functions with humans at various levels. For example, the human acetyl transferases Gcn5 and Tip60 and the methyl transferase SetD2 have conserved counterparts in yeast [88,89,90]. A notable exception is the PRC2 complex, which methylates H3K27 in humans, and is not present in *S. cerevisiae* or *S. pombe* [91]. While many histone modifying enzymes are conserved, there are limitations regarding which ones can be investigated in yeast models as some mammalian proteins lack yeast orthologs. In addition to conserved associated enzymes, the functional consequences of many key PTMs are similar between yeast and mammals. For example, H3K36 methylation promotes double-strand break repair in budding yeast and humans [82,92,93], and heterochromatin is marked by methylation at H3K9 in fission yeast and humans [52,94]. There are differences, though, with both budding and fission yeast lacking H3K27 methylation [95]. Thus, yeast systems can model the placement and consequence of many, though not all, epigenetic marks found in mammalian species.

The yeast model systems offer flexibility in how to analyze the functional consequences of oncohistone mutations by examining cells that express the oncohistone protein as the sole copy of the histone present. In a mammalian cell line, engineering the 15 copies of histone H3 to express the oncohistone as the sole histone protein for clean characterization of its impact is quite technically challenging, although this task has been accomplished for H3K27 [87]. However, in the budding yeast system, where each histone is encoded by only two genes, one can readily engineer cells that express an oncohistone as the sole histone protein present. In fact, offering flexibility, one copy of a histone gene can be altered using genome editing and then the other histone gene can be kept intact or deleted (Figure 4A). Studies can then be performed using cells where a single allele is edited and the other allele is wild type, for example *hht2-K36M HHT1*, to more closely model what happens in cancer cells where the oncohistone allele is dominant to the numerous wild type alleles of any histone gene (Figure 4A). Alternatively, if the locus encoding the second histone gene is deleted, this creates a model where the oncohistone variant is the sole histone protein expressed in cells (for example *hht2-K36M hht1Δ*) (Figure 4A). This latter experimental design allows direct analysis of how a specific oncohistone mutant alters histone function. These two scenarios provide the flexibility to assess (1) dominant phenotypes in the presence of the other wildtype histone gene, which aligns with what happens in oncohistone-driven cancers where only a single allele among many is altered and (2) recessive phenotypes in the absence of any wildtype histone gene. The ease of generating these models allows for the rapid screening of a large number of potential oncohistone mutations to assess whether they are likely to alter histone function.

Another advantage of the yeast model system is the ability to screen for numerous phenotypes by plating oncohistone models on plates under a variety of conditions, including different temperatures, nutrient sources, and drugs that disrupt different pathways [96] (Figure 4B). Such growth assays have been exploited extensively to systematically analyze the functional importance of different histone residues for specific PTMs and cell fitness [97,98,99,100]. As an example of this modeling, we performed a serial dilution assay of the oncohistone H3K36M and the mutant H3K36R, which has been identified in T-cell acute lymphoblastic leukemia, though its oncogenicity is uncertain [101], and is expressed both in the presence of wildtype H3 (*hht2-K36M/R HHT1* cells) and as the sole copy of H3 (*hht2-K36M/R hht1Δ*) (Figure 4B). To illustrate this experimental approach, results of such an experiment are presented in Figure 4B. For this experiment, yeast cells were grown in culture, serially diluted, and plated on control plates, plates that contain caffeine, which induces cellular stress [102], or plates that contain phleomycin, which induces double-strand breaks [103]. H3K36M cells show sensitivity to cellular stress and double-strand breaks at fairly similar levels in the presence or absence of wildtype proteins. On the other hand, H3K36R cells show an increase in sensitivity to cellular stress and double-strand breaks when H3K36R is expressed as the sole copy of histone H3 (Figure 4B). Analyses such as these can provide insight into the genetic and biochemical properties of oncohistone mutants. 

Yeast oncohistone models are valuable for extending such an analysis to explore specific amino acid changes identified in cancer patients, like those displayed in Figure 3, to identify mutations that alter histone function. For example, a previous investigation assessed the effects of cancer-associated core histone mutants on chromatin remodeling (104). A humanized yeast library was created and employed to discover that certain histone mutants increase histone exchange and nucleosome sliding [104]. Additionally, when these histone mutants were expressed in mammalian cells, it was discovered that cancer-associated gene pathways were upregulated [104]. This investigation is a helpful example of how the power of yeast genetics can be exploited for the rapid discovery and translational studies of human disease-related mutants.

Given the ease of genetic manipulation and unbiased screens in yeast, one might consider expressing a fully humanized nucleosome in the model system. A previous study deleted the core histone genes in *S. cerevisiae* and performed a plasmid shuffle to express only the human histone proteins [105]. This process occurred at very low rates, and cells accumulated many suppressor mutations to accommodate growth. Altering two residues in H3 (P121K, Q125K) and three residues in H2A (Q113H, A114Q, V115N) back to the *S. cerevisiae* sequence greatly improved efficiency of “humanization”. Interestingly, humanized nucleosomes revealed similar spacing between nucleosomes to that found in wildtype yeast and caused an overall reduction in total RNA production [105]. Although the histone protein sequences are very similar between humans and budding yeast, these differences, in combination with the species–specific epigenetic factors and chromatin remodelers, may be sufficient to cause the phenotypes observed in the humanized strain. Certain experimental questions may be best answered with budding yeast which expresses fully humanized nucleosomes, but the consequences of this expression on chromatin and RNA production would need to be considered.

## 5. Progress towards Histone and Oncohistone Characterization via Yeast Models

Epigenetic discoveries in yeast have contributed to our understanding of the chromatin-driven regulation of gene expression. Specific examples include modulators of histone acetylation and their role in regulating transcription. The deacetylase Rpd3 (human HDAC3) was originally identified in budding yeast through a screen designed to identify transcriptional regulators, which then enabled the discovery of the related deacetylase in mammalian cells [106,107]. The acetyl transferase Gcn5 (human GCN5/KAT2A) was also first discovered and investigated in budding yeast [108,109]. Moreover, characterization of the Swi-Independent (SIN) histone mutants helped elucidate the fact that the SWI/SNF complex is critical for chromatin remodeling and revealed histone residues that are involved in DNA–protein interactions [110]. Additionally, the power of yeast genetics is evident in screens that explored the critical roles played by specific histone residues. The results from these screens proved to be foundational to our current understanding of histones. Tandem alanine mutants in histone H3 revealed specific sensitivities to DNA damaging agents and that the αN helix is crucial for its chaperone function, particularly for nucleosome assembly and disassembly [111]. Another study that examined alanine or PTM mimetic mutants of H3 and H4 provided insight into which histone residues influence chronological lifespan [112]. This analysis identified various residues and epigenetic marks that either extended or reduced lifespan which may correlate with destabilizing histone–DNA or histone–histone interactions [112]. An additional investigation discovered surface-accessible residues in all of the core histone proteins that are critical for DNA-interacting functions, such as transcription initiation, transcription elongation, and DNA repair [100]. Furthermore, alanine screening of H4 revealed three adjacent C-terminal residues (L97, Y98, and G99), all of which are conserved in humans, that are required to protect cells from genome instability and ensure proper histone occupancy across the genome [113]. These systematic epigenetic investigations performed in budding yeast helped to provide a foundational understanding of histone protein structure and function, and the resulting discoveries propelled mammalian epigenetic research into an analysis of disease-relevant mutations, such as oncohistones. Thus, there is great power in employing model organisms to explore mutational landscapes, including oncohistone mutants.

Despite the fact that yeast does not develop cancer, taking advantage of these approaches has revealed altered growth patterns associated with yeast oncohistone models. In budding yeast, the expression of either H3K36M or H3K36R as the sole copy of H3 results in reduced growth in the presence of caffeine, suggesting the sensitivity of these oncohistone models to cellular stress [114]. This caffeine sensitivity is also observed when methylation at H3K36 is prevented by deleting the *SET2* gene which encodes the H3K36 methyltransferase [40]. Notably, yeast Set2 also binds to H3K36M, as does human SETD2 [90]. As described above, one of the greatest strengths of using yeast as a model system is the ability to perform unbiased screens. To identify factors and pathways upon which H3K36 mutants are specifically sensitive, our group took advantage of the caffeine-sensitive growth of H3K36 mutant cells in the absence of wild type protein to perform a high copy suppressor screen [114]. The goal of such a screen is to identify suppressors that may identify therapeutically actionable pathways in the corresponding cancers. We identified various suppressors of growth defects on caffeine [114], some of which have known roles in regulating histone function, such as the lysine acetyltransferase Esa1 [115], and a putative transcriptional regulator that interacts with Rpd3 and Set3 histone deacetylase complexes, Tos4 [116]. Further experimentation to elucidate the mechanism of suppression has the potential to provide insight into mechanisms that underlie altered histone function in cancer. 

Many other investigations of oncohistone mutants in yeast have provided insight into the physiological changes that could contribute to oncogenesis. H3G34R mutants in fission yeast result in reduced levels of methylation and acetylation at H3K36 [67]. H3G34V mutants in fission yeast reveal sensitivity to induced DNA double-strand breaks, while H3G34R mutants are sensitive to DNA replication stress and defective in homologous recombination [78,117]. These results are congruent with what has been found in humans: that cells expressing mutant H3G34 display heightened frequency of genomic mutations [69]. H3K27 methylation is absent in *S. cerevisiae* and *S. pombe*, possibly because their mechanisms for inhibiting gene transcription are distinct from the heterochromatin present in mammalian systems [91]. Acetylation at this site does occur in yeast, however [118]. Due to this inconsistency with human epigenetic regulation, as well as the absence of the PRC2 complex as described above, yeast has not been employed as a model for the H3K27M oncohistone. 

The strengths of employing yeast models to characterize novel oncohistones are also exemplified through investigations of putative oncohistones and characterizing possible oncohistone mechanisms. The variant H2BE76K, which occurs in bladder and head and neck cancers, is found in the globular domain of histone H2B [76]. In budding yeast, engineering one H2B gene to harbor the H2BE76K mutation while maintaining one wildtype copy causes temperature sensitivity and reduced nucleosome stability [76]. Additionally, an array of variants in histone residues located in the acidic patch or histone–DNA interface that are common in cancers displayed increased chromatin remodeling processes and lethal growth defects in fission yeast, suggesting that cancer-related mutants in these residues could also be investigated in the model organism [104]. Finally, as many cancer-associated mutations result in lysine-to-methionine variants (K-to-M), understanding the mechanism by which these changes disrupt cellular growth is crucial. The H3K9M mutant in fission yeast impaired global methylation at H3K9, which is found in constitutive heterochromatin, and displayed enhanced interactions with the human methyltransferase G9a [119]. This mechanism of a K-to-M mutant resulting in tighter association with its respective modifying enzyme is similar to what is observed in the H3K27M and H3K36M oncohistones [61,62,68,70]. Thus, yeast can be employed to model this mode of oncohistone mutant. Yeast model organisms have proven to be useful not only for initial fundamental histone investigations, but also for the ongoing characterization of clinically relevant oncohistone mutants. The genetic simplicity of these models, the ability to readily screen for phenotypes, and the power of genetic screens will continue to provide insight into disease etiology that can be readily extended to humans.

## 6. Conclusions

Studies in yeast models can be complementary to studies in human cells to define mechanisms by which oncohistones subvert the key function of histones. In recent years, budding yeast oncohistone models were used in a screen to identify potentially therapeutically actionable suppressors of growth alterations [114], and the impact of oncohistone mutants on DNA repair was characterized in fission yeast [78]. Analyzing candidate oncohistones, here we present recurrent changes that occur in amino acids conserved between human H3.3 and budding yeast, which may be worthy of further investigation to assess their contribution to disrupted growth patterns. We also provide a comprehensive comparison of gene number, protein variants, and protein sequences between human and yeast histones that should be a valuable resource for the scientific community. In summary, yeast serves as an excellent model for the analysis of oncohistone mutants, providing a simple system to assay function either in the presence or the absence of wild type histone protein, which is technically demanding in a mammalian system. The findings revealed in yeast will drive the field forward, enabling molecular insights to initiate translational studies and, hopefully, lead to improved treatment paradigms for patients with oncohistone-driven cancers.

## Figures and Tables

**Figure 1 jof-09-01187-f001:**
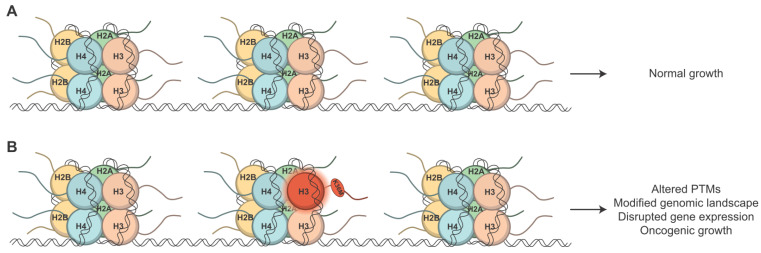
Nucleosome structure and the consequences of an oncohistone mutant. (**A**) Nucleosomes are comprised of two copies of each of the four core histone proteins: H2A, H2B, H3, and H4. When only wild type histones are present, cells display normal growth. (**B**) When an oncohistone missense mutation occurs, such as H3K36M (red), various functional consequences may result. Even with many wildtype H3 proteins present, a single mutation in one allele can confer oncogenic growth through a variety of mechanisms, including altered post translational modifications, modification of the genomic landscape, and/or disruption of gene expression.

**Figure 2 jof-09-01187-f002:**
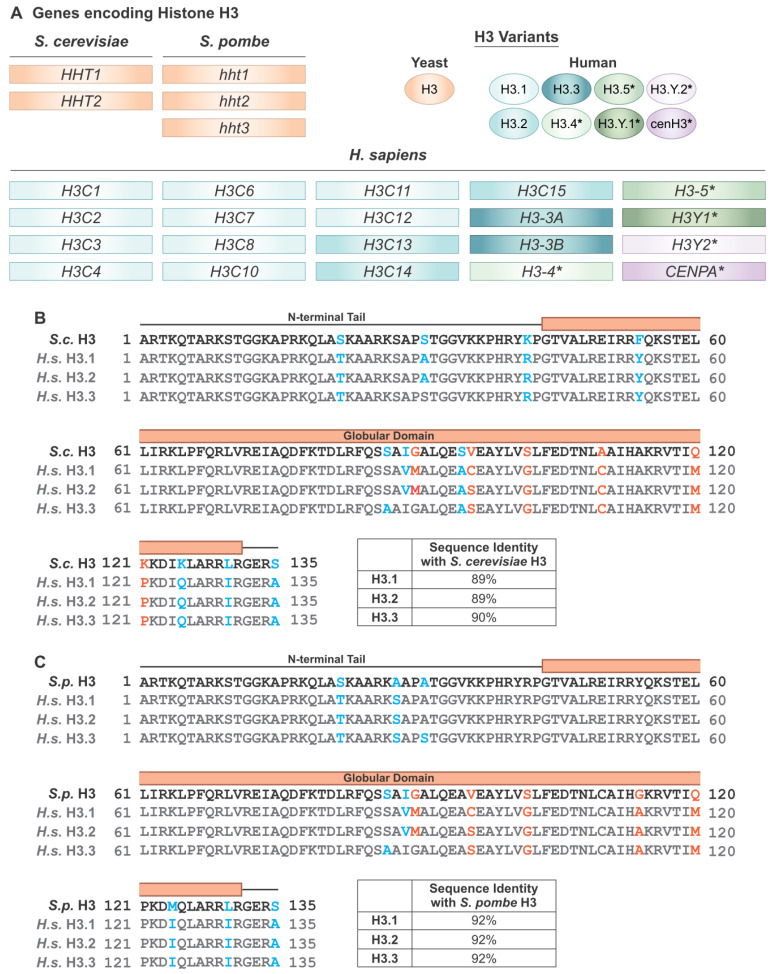
A comparison of histone H3 from *S. cerevisiae*, *S. pombe*, and *H. sapiens*. (**A**) The number of gene copies and protein variants differs greatly between yeast species (*S. cerevisiae* and *S. pombe*), which are termed *HHT* genes, and humans (*H. sapiens*), which are termed H3XX, with the exception of the specialized centromeric histone CENPA for histone H3. Nomenclature guidelines for each species were followed. The genes that encode each variant are color coordinated with their respective H3 Variants. * Non-canonical H3 genes and variants. (**B**) Protein sequence alignments for histone H3 in *S. cerevisiae* and the three canonical H3 variants in *H. sapiens*. Blue residues represent conservative changes, where the biochemical properties of the amino acid are maintained, and orange residues represent non conservative changes, where the biochemical properties are altered. (**C**) Protein sequence alignments for histone H3 in *S. pombe* and the three canonical H3 variants in *H. sapiens*. Color coding is the same as for (**B**).

**Figure 3 jof-09-01187-f003:**
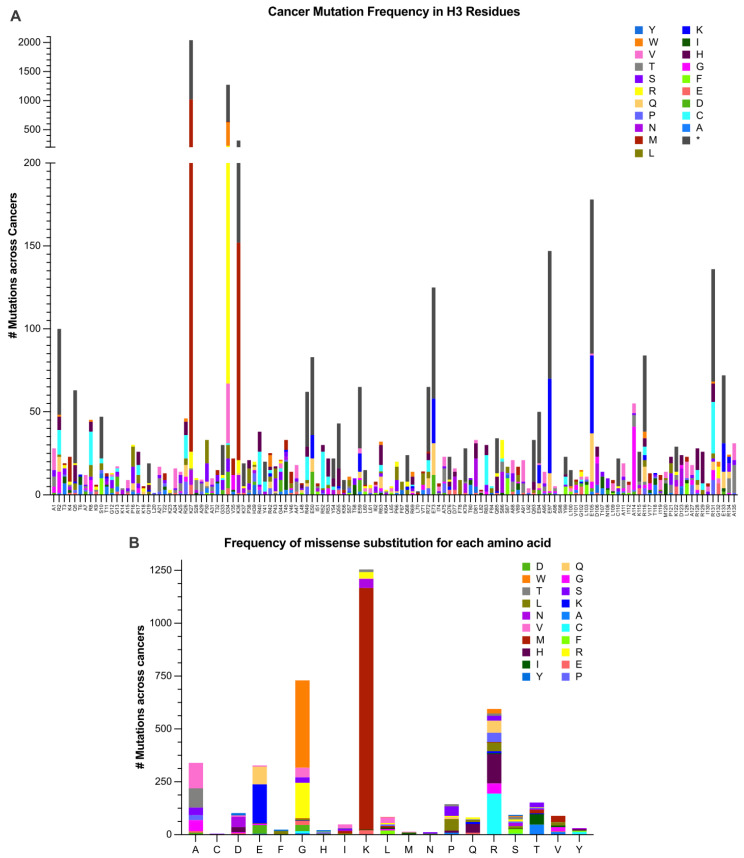
Mutations that alter histone H3 sequence recur in human cancers. (**A**) A cross-cancer mutation analysis was performed utilizing 5406 adult patient samples obtained from the COSMIC and cBioPortal databases. Samples from the two databases were cross-referenced and duplicate entries were removed. The data were then visualized using PRISM GraphPad. This summary includes missense mutations found in H3C1, H3C2, H3C3, H3C4, H3C5, H3C6, H3C7, H3C8, H3C10, H3C11, H3C12, H3C13, H3C14, H3C15, H3-3A, H3-3B, and H3-4. The number of patients that have a mutation (missense or premature termination codon as indicated by *) located in the codon corresponding to each specific amino acid residue along the protein is indicated on the Y-axis (# Mutations across Cancers) with the residues that the amino acid is altered to being indicated by the colors or the asterisk (*). Only residues that are conserved in *S. cerevisiae* H3 are shown along the X-axis as these represent potential oncohistones that could be modeled in budding yeast. (**B**) The number of events for which a given amino acid is converted into all other amino acids was assessed, regardless of position in the protein sequence. Premature termination codons were excluded to focus on missense mutants. Same dataset as in (**A**).

**Figure 4 jof-09-01187-f004:**
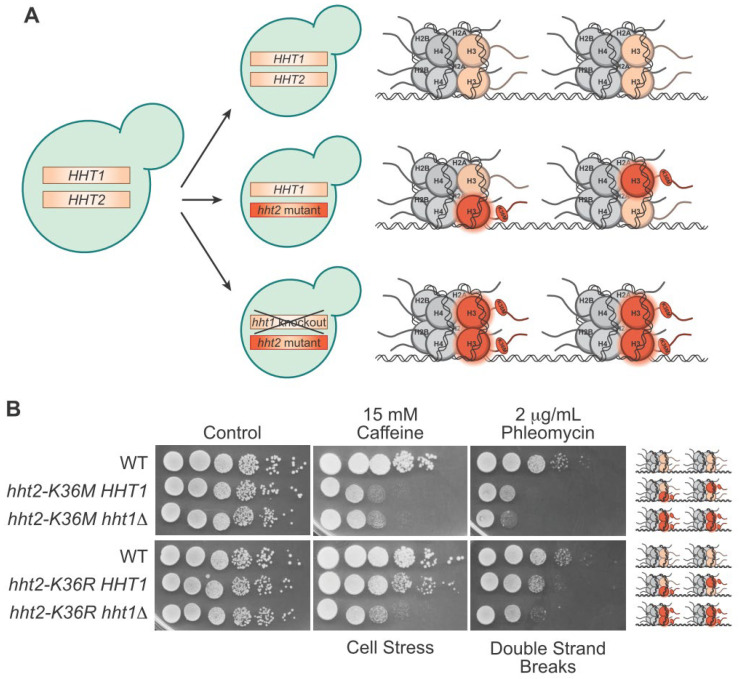
The strength of modeling oncohistone mutants in budding yeast. (**A**) Budding yeast can be easily engineered to maintain wildtype H3 genes (**top**), express one mutant and one wildtype H3 (**middle**), or express only mutant H3 with the wildtype knocked out (**bottom**). The resulting ratio of wildtype to mutant H3 proteins in a pair of nucleosomes for the given genotypes is depicted on the right. (**B**) In this serial dilution growth assay, budding yeast cultures were diluted to OD = 5 and serially diluted tenfold before plating onto control or drug plates. The analysis compares H3K36 mutant cells that either have one mutant and one wildtype histone protein (*hht2-K36M/R HHT1*) and cells that contain mutant histone as the sole copy of histone H3 (*hht2-K36M/R hht1Δ*) to control wildtype (WT) cells. Cells on control plates were grown for two days, cells on caffeine were grown for five days, and cells on phleomycin were grown for three days. The cellular damage caused by the drugs is indicated below the plates, and the ratio of wildtype to mutant H3 in the nucleosomes within each model is depicted to the right.

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
