# Peer review of "Exploring the Molecular Underpinnings of Cancer-Causing Oncohistone Mutants Using Yeast as a Model"

_jof, 2023, doi:10.3390/jof9121187_

Round 1
Reviewer 1 Report
Comments and Suggestions for Authors
The authors propose to investigate/review some interesting histone questions:: 1) Why do humans have so many histone genes and histone variant proteins compared to the simplicity of yeast? 2) Do subtle differences in the human histone proteins really translate to essential functional differences? 3) Can yeast histone knowledge help us understand human histone variants, what they do/why we need them? These are only partially addressed in this review article. In addition, the article fails to consider some key literature regarding the characterizations of histones, nucleosomes and oncohistone mutants in yeast. There is too much emphasis on human nucleosomes/epigenetics (that is already extensively reviewed by others) without comparing contrasting to yeast epigenetics. The review would be greatly strengthened with the inclusion and discussion of additional research.
Specifically,
1. I think there is an opportunity lost in the overview of nucleosomes and histones section to compare and contrast histones/nucleosomes from yeast with human:
a. Add reference for crystal structure of yeast nucleosome and histone octamer in yeast and some text describing how yeast nucleosomes are spaced closer together, with a repeat length of 162 ± 6 bp (Horz and Zachau, 1980), resulting in a linker length of only 15–20 bp.
b. Does the linker histone h1 function differently in yeast? –does yeast not require an H1 to the same extent as humans due to differences in the molecular surface of the yeast octamer? What Are the s cerevisiae and s pombe H1 genes and are there additional proteins with this function?
c. Yeast nucleosomes unwind faster with heat (Morse 1987) – so the yeast histone octamer may be less stable than the human histone octamer? I believe the yeast nucleosome crystal structure also supports this by showing that the interaction between H2A and H2B is much weaker in yeast than humans.
d. Nucleosome packing may be different in yeast? Would you agree that is true?
e. In general, the yeast genome is constitutively “open” for transcription, as opposed to the small percentage of actively transcribed genes at any given time in humans. Thus, there may be fundamental differences in mechanisms that maintain heterochromatin in yeast since they aren’t needed to the same degree as in humans.
2. Yeast H2A and H2B are only about 70% homologous to humans yet histone octamer is functionally and structurally about the same. So why do humans need all these extra histone genes?
3. Take a look at Truong and Boeke’s 2017 paper where they humanized the yeast epigenome by replacing yeast H3 with human H3.1. They found 5 amino acids were important to convert the human nucleosome to allow yeast growth. Papers like this would be interesting to include/discuss in this review.
4. Is it correct to state the histone code is well conserved between yeast and humans? Maybe highlight some similarity and differences between yeast and human in the paragraph that starts on line 99. For instance, I’m not sure it is correct to say that methylation of histone tails is conserved in yeast since S. cerevisiae and S. pombe don’t have PRC2 and consequent H3K27 methylation so I’m not sure yeast would be the best model to study K27M.
5. Are the two h3 genes in cerevisiae and 3 in pombe expressed throughout the cell cycle or is their expression limited to S phase? Also, is there 100% homology between the protein product of yeast genes (only one protein product)?
6. Does H3K36M bind Set2 in yeast (similar to mechanism of SETD2 in humans)? (line 186) How can this mechanism be similar in yeast if there is no consequent redistribution of H3K27me3? (line 207) Does yeast Set2 also bind G33-34?
7. Your cBioportal/COSMIC analysis had 5400 patient samples and it looks like about 2000 (40%) had a H3K27 mutation in Fig 2A? That seems like a high frequency for mutation at that site across all cancers. Is this due to some bias in the sample sets chosen? But then at line 264 in the text you state there were 1215 instances of missense mutation at H3K27 so why is the bar about 2000 on the graph in Fig 3A?
8. The premise for the review is that yeast models are a good resource to study whether histone mutations found in cancer databases are passenger or driver mutations. What is the functional significance of those yeast histone H3 mutations that are analogous to the human mutations found in cBioportal? Perhaps add some discussion of the results in Bagert et al 2021.
9. Why do HHT2 and HHF2 contribute more than 80% of the mRNA transcripts?
10. Is it known why deletion of HTA1- HTB1 is lethal while HTA2-HTB2 deletion is tolerated? Are A1-B1 responsible for most transcripts? Are these budding yeast histone genes expressed throughout the cell cycle or restricted to a specific cell cycle phase?
11. Please expand the section that begins at line 308 – I would love to see more detail for how the initial characterization of yeast histone epigenetics laid the groundwork for mammalian studies.
12. Line 311 – It would be useful to add the name of the human paralog protein – For instance, I think Rpd3 is HDAC3 in humans and it would be useful to note that.
13. Line 317 – Are there databases online with the alanine scanning/dna damage screen results (does histonehits.org or some update still exist somewhere?)? It would be useful to provide links if so.
14. Line 325 - please list the 3 residues, are they also present in human?
15. Please add some discussion of SIN mutants and chromatin remodelers. I don’t think the review is complete without some discussion of SIN mutants.
16. Line 334 please list some specific disadvantages to studying these mutants in human cell lines. line 339-340 – Most histone mutants in human cell lines will only account for about 5-10% of total histone yet these mutants still drive a phenotype. Won’t mutants in yeast histone genes account for a much higher frequency of the total histone pool e.g. 80-90% of histones will be mutant. For instance in human cells what % of total histone ins H3K36M and what % is it when expressed in yeast, are they similar?
17. Line 350-351 – Although the amino acid sequence of human and yeast H3 N-terminal tails are almost identical, yeast may lack the same epigenetic mechanisms to modify these residues or they may differ in how they regulate transcription and so these amino acids may not serve similar biochemical functions. Can you provide more evidence that well characterized epigenetic marks in humans have a similar function in yeast. E.g. is K27Ac associated with active enhancers? It is my understanding that S. cerevisiae lack the PRC2 mechanism so K27 methylation is a major difference between budding yeast and humans.
18. Line 356 – 357: “These similarities suggest that histone H3 performs analogous functions in each species and, thus, that molecular disruptions caused by the oncohistone mutants would be similar.” Does K27M really have a function in budding yeast that is analogous to humans? Please expand your review to include discussion of key results from studies of the well-known oncohistone mutations such as K27, G34, K36 – what are the functional consequences of expressing these in yeast?
19. Line 384 – It wouldn’t be physiologically relevant to generate cells with the oncohistone being the sole histone gene expressed – I don’t think it should be a goal to generate such cells in humans.
20. Line 462-463: Please include some text describing how yeast can be leveraged to study nucleosome positioning defects of histone mutants in stress-induced systems such as the amino acid induced yeast system described in Bennett et al 2019 where they found a particular nucleosome was less likely to occlude the PHO5 promoter with H2B was mutant. This was a missed opportunity to describe this here.
Reviewer 2 Report
Comments and Suggestions for Authors
In this manuscript, Zhang and colleagues review the emerging area of oncohistones (i.e., histone mutations that drive oncogenesis). The authors propose that, due to the conservation and simplicity of their genomes, both budding and fission yeasts can be good models for characterizing oncohistones. In addition, the authors summarize prior work investigating yeast histones and present some of their own data illustrating the power of yeast model systems in this field. I recommend this article for publication with the following modifications.
Figures 2 and S1:
The authors state “Orange residues represent conservative changes, where the biochemical properties of 138 the amino acid are maintained, and blue residues represent non conservative changes…” However, it is the other way around. In addition, S. pombe genes should be written in lowercase italics (e.g. hht1).
Paragraph 99-130:
The authors should mention H3K9 and H3K27 as good examples of methylation as they are hallmarks of heterochromatin and transcriptional silencing.
Paragraph 446-458:
While the authors clarify that H3K27me is not present in yeasts, they should mention that H3K9me exists in S. pombe and cite work showing that mutations such as K9M (Shan et al.; Elife 2016 PMID:27648579) and G13D (DiPiazza et al.; PNAS 2021 PMID:34035174) can impair methyltransferase activity, as seen with the highly frequent K27M oncohistone.
Figures 1 and 4:
Resolution is very poor. It is not possible to read the small fonts. On the other hand, can the authors try a lower phleomycin concentration (e.g. 1 µg/ml), which would allow cells to grow more and make the difference more obvious?
Minor issues
Lines 365: Italics missing.
Lines 415-416: “(ref)”?
Comments on the Quality of English Language
Line 332: Colloquial language: “a little over a decade ago”
Round 2
Reviewer 1 Report
Comments and Suggestions for Authors
The authors have been very responsive to our suggestions and the manuscript is a substantial contribution
I have a few minor comments to address before publication
1- The genes for canonical H2B and H2A encode proteins that have a few amino acid changes, almost none of the H2A or H2B proteins are identical - hence there are more isoforms of H2A and H2B than indicated in the figures; The H2A and H2B variants are specialized and often encoded outside the histone gene clusters. So there are actually more than 10 H2A and 4 H2B proteins when you count the proteins encoded by the canonical genes. There are indeed only 3 types of H3 (replication dependent H3.1 and 3.2 and replication independent H3.3) but the H4 genes encode 3 types of H4 which vary only by one or two amino acids. Studies in which the variable aa sequences of the canonical histone proteins, if inserted into yeast, might reveal the subtle differences in histone protein function- this should be pointed out.
2- Page 5 top- the histone linker length is not 160-200 bp in mammals- that length I believe refers to the nucleosomal repeat length - please correct and either state the internucleosomal repeat length is smaller in yeast than mammals and that the the linker length between the nucleosomes is longer in mammals than yeast
Here is a quote from a 2001 Luger review paper
"Yeast nucleosomes are very closely spaced, with a repeat length of 162 ± 6 bp (Horz and Zachau, 1980), resulting in a linker length of only 15–20 bp. In contrast, the repeat length in metazoans ranges from 175 to 240 bp, with an average of ∼190 bp. "
From this you can infer that the linker length in metazoans is in the range of 40-100 bp
3- non sense codons might be important- this could cause to a deficiency of histone and upset the balance for complete histone octamer formation; studies of loss of one allele of the yeast histone genes might be informative in this regard.
Reviewer 2 Report
Comments and Suggestions for Authors
Manuscript has been improved and it is now suitable for publication.
Author Response
Thank you.